# Genetic Diversity and Possible Origins of the Hepatitis B Virus in Siberian Natives

**DOI:** 10.3390/v14112465

**Published:** 2022-11-07

**Authors:** Victor Manuylov, Vladimir Chulanov, Ludmila Bezuglova, Elena Chub, Anastasia Karlsen, Karen Kyuregyan, Yulia Ostankova, Alexander Semenov, Ludmila Osipova, Tatjana Tallo, Irina Netesova, Artem Tkachuk, Vladimir Gushchin, Sergey Netesov, Lars O. Magnius, Heléne Norder

**Affiliations:** 1Gamaleya National Research Center of Epidemiology and Microbiology, 123098 Moscow, Russia; 2National Medical Research Center for Phthisiopulmonology and Infectious Diseases, 127473 Moscow, Russia; 3Chair of Infectious Diseases, Sechenov University, 119048 Moscow, Russia; 4Hepatitis B ELISA Department, Vector-Best JSC, 630559 Koltsovo, Russia; 5Department of Molecular Virology of Flaviviruses and Viral Hepatitis, State Research Center of Virology and Biotechnology “Vector” of the Rospotrednadzor, 630559 Koltsovo, Russia; 6Department of Viral Hepatitis, Russian Medical Academy of Continuous Professional Education, 125993 Moscow, Russia; 7Laboratory of Viral Hepatitis, Mechnikov Research Institute of Vaccines and Sera, 105064 Moscow, Russia; 8Scientific and Educational Resource Center for High-Performance Methods of Genomic Analysis, Peoples’ Friendship University of Russia (RUDN University), 117198 Moscow, Russia; 9Laboratory of Molecular Immunology, Institute Pasteur in Saint Petersburg for Research in Epidemiology and Microbiology of the Rospotrednadzor, 197101 Saint-Petersburg, Russia; 10Ekaterinburg Research Institute of Viral Infections of SRC VB Vector, 620030 Ekaterinburg, Russia; 11Laboratory of Populational Ethnogenetics, Department of Molecular Diagnostics and Epidemiology, Institute of Cytology and Genetics, 630090 Novosibirsk, Russia; 12Department of Microbiology, Public Health Agency of Sweden, 171 82 Stockholm, Sweden; 13Laboratory of Bionanotechnology, Microbiology and Virology, Novosibirsk State University, 630090 Novosibirsk, Russia; 14Ulf Lundahl’s Foundation, 100 61 Stockholm, Sweden; 15Department of Infectious Diseases, Institute of Biomedicine, University of Gothenburg, 413 90 Gothenburg, Sweden; 16Department of Clinical Microbiology, Sahlgrenska University Hospital, 413 45 Gothenburg, Sweden

**Keywords:** hepatitis B virus, genotypes, subgenotypes, HBsAg subtypes, molecular epidemiology, Siberia, Siberian natives, aboriginal population

## Abstract

A total of 381 hepatitis B virus (HBV) DNA sequences collected from nine groups of Siberian native populations were phylogenetically analyzed along with 179 HBV strains sampled in different urban populations of former western USSR republics and 50 strains from Central Asian republics and Mongolia. Different HBV subgenotypes predominated in various native Siberian populations. Subgenotype D1 was dominant in Altaian Kazakhs (100%), Tuvans (100%), and Teleuts (100%) of southern Siberia as well as in Dolgans and Nganasans (69%), who inhabit the polar Taimyr Peninsula. D2 was the most prevalent subgenotype in the combined group of Nenets, Komi, and Khants of the northern Yamalo-Nenets Autonomous Region (71%) and in Yakuts (36%) from northeastern Siberia. D3 was the main subgenotype in South Altaians (76%) and Buryats (40%) of southeastern Siberia, and in Chukchi (51%) of the Russian Far East. Subgenotype C2 was found in Taimyr (19%) and Chukchi (27%), while subgenotype A2 was common in Yakuts (33%). In contrast, D2 was dominant (56%) in urban populations of the former western USSR, and D1 (62%) in Central Asian republics and Mongolia. Statistical analysis demonstrated that the studied groups are epidemiologically isolated from each other and might have contracted HBV from different sources during the settlement of Siberia.

## 1. Introduction

Human hepatitis B virus (HBV) belongs to the *Hepadnaviridae* family. It has a circular DNA genome of ~3200 nucleotides encoding four major proteins: P, S, C, and X. Differences in the structure of the surface antigen HBsAg, encoded by the S-gene, have allowed researchers to identify nine HBV subtypes: ayw1, ayw2, ayw3, ayw4, ayr, adw2, adw4, adrq+, and adrq− [1,2,3]. The amino acid residues that determine these subtypes have been described [4,5]. Phylogenetically, HBV is classified into at least eight genotypes designated A through H [6,7,8,9,10,11,12]. Some authors additionally distinguish genotype I, which is considered a recombinant between genotypes A, C, and G, and the rare genotype J, which is highly divergent from others human HBV strains and is close to a gibbon HBV [13,14]. Furthermore, within the genotypes, at least 42 subgenotypes have described, designated with Arabic numerals (A1, A2, etc.) [15,16,17,18,19]. Strains of the same HBsAg subtype may belong to different (sub)genotypes and vice versa, but there is a predominant correlation between serological subtypes and genotypes [14,17]. The subtype and genotype classifications of HBV complement each other and are used in serological research (e.g., studies on vaccine efficacy or serological diagnostics), as well as in genetics studies on the evolution and molecular epidemiology of HBV.

The prevalence of different HBV (sub)genotypes and HBsAg subtypes varies in geographical regions [3,17,20]. Genotype A (subtypes ayw1 and adw2) dominates in northwestern Europe, North America, Africa, and Asia, with subgenotype A1 common in Africa and Asia and A2 in Europe and the USA. Genotypes B (subtypes ayw1 and adw2) and C (subtypes adr, adrq+/−, ayr, and adw2) predominate in southeastern Asia and Oceania. Genotype D is the most widely distributed genotype, but its main subgenotypes vary in different parts of the world: D1 (ayw2) predominates in the Middle East and in Turkic and Maghreb states; D2 (ayw3) is common in Eastern Europe; and D3 (ayw2) circulates as a minor variant in South America (cited from [17,20]). In some regions, such as the Mediterranean, India, and Russia [21,22,23,24,25], these three D-subgenotypes are intermixed and circulate together. Genotype E (subtype ayw4) is prevalent in the East African countries, and genotypes F and H (subtypes adw4 and ayw4) are the main genetic variants in South and Central America [12]. Several isolates of genotype G have been described in the USA, Western Europe, and Asia; genotype I is common from India to Vietnam; and some strains of genotype J have been detected in Japan [11,13,26,27,28,29,30].

The Siberian region, which traditionally includes territories from the Russian Far-East to Eastern Ural, covers around 13 million square kilometers (~75% of the Russian Federation) and has ~40 million inhabitants (more than 25% of RF’s total population). Siberia is inhabited with unique indigenous population groups that have historically lived in geographic and cultural isolation from each other and from the Russian settlers. Some Siberian natives have been assimilated by newly arrived populations and now reside in cities, but many still live in remote settlements, preserving their traditional lifestyle. Due to their isolation, local native groups may be epidemiological reservoirs in which HBV and other pathogens have persisted and evolved for a relatively long time after their introduction without intermixing with strains from other populations. If this is true, analyzing these isolated HBV populations may help to disclose how HBV historically spread throughout Siberia. In this study, we aimed to investigate the prevalence of HBV subgenotypes and HBsAg subtypes in nine groups of native Siberian populations and also to deduce historical origins of the revealed pattern of HBV genetic diversity in Siberia.

## 2. Materials and Methods

### 2.1. HBV Strains

A total of 381 HBV DNA sequences collected from 9 groups of native Siberian populations (Figure 1) were analyzed along with 179 HBV strains sampled from different urban populations of the western republics of the former USSR (including Russia) and 50 strains from the former Central Asian USSR republics and Mongolia (Table 1). Some of these sequences were collected by the authors of this paper, while others were retrieved from GenBank (see Table 1 for the references and accession numbers). Also included in the analysis were 124 HBV strains from ancient tombs located in different parts of the world (including modern Russian and Central Asian territories), which were sequenced by [31]. Additionally, 319 prototype strains of different HBV genotypes and subgenotypes were used for general subgenotyping analysis. For them, GenBank accession numbers and countries are listed in Appendix A. 

The analyzed sequences varied in length from 681 nucleotides to 3225 nucleotides. Even the shortest sequences included the entire sequence of the most variable HBV S-gene (nucleotide positions 1564–2244 according to the prototype isolate X02763 [32]). Alignment of all the studied sequences with indicated accession numbers or unique codes, country or region of origin, and year of collection is attached in Appendix A in PHYLIP and FASTA formats. All the listed data for the strains (country/city, population group, GenBank accession number, year of collection) are provided in Appendix A. 

### 2.2. Phylogenetic Analysis

Phylogenetic analysis was performed using maximum likelihood (ML) method with approximate likelihood ratio test for branches based on the Shimodaira-Hasegawa procedure (aLRT-SH) [33] in the online version of PhyML 3.0 software [34], http://www.atgc-montpellier.fr/phyml/. For the analysis, the following parameters were used: general time reversible (GTR) substitution model; empirical equilibrium frequencies; estimated proportion of invariable sites; 4 substitution rate categories; gamma shape parameter estimated; and aLRT-SH-like branch support method. A strain was assigned to a specific HBV subgenotype if it was clustered into the supported (index ≥ 90) branch corresponding to this subgenotype. The indices are shown in Figure 2 and Appendix A.

### 2.3. Determination of HBsAg Subtypes

HBsAg subtypes were determined by analyzing amino acids residues at positions 122, 127, 140, and 160 encoded by the S-gene of the HBV genome, as previously described [5].

### 2.4. Statistical Data Processing

To perform pairwise comparisons for the proportions of the HBV variants in the studied groups, 2-sided Fisher’s exact test or Chi-square criterion with Yates correction were used (depending on sample characteristics). The significance threshold was set at *p* < 0.05; specific *p*-values are provided in the text.

**Table 1 viruses-14-02465-t001:** List of the HBV strains included in the study (n = 1053).

Group and Region	No. of Strains	Year(s) of Collection	GenBank Accession Numbers	Citation
Siberian Natives, n = 381				
Altaians: Altai Republic, Ust-Kansky District (southwestern Siberia)	21	1997	JX090656-JX090674, JX125378, JX125379	[25]
Altaian Kazakhs: Altai Republic, Kosh-Agachsky District (southwestern Siberia)	7	1999	JX090719-JX090724, JX125377	[25]
Tuvans: Tyva (or Tuva) Republic (southern Siberia)	18	2004–2008	Author’s numbers (tuvXX, see Appendix A)	[35]
Teleutes: Kemerovo Region, Belovsky District (southwestern Siberia)	5	2003	JX090633-JX090637	[25]
YNAR: combined group of Khants, Nenets, Komi, Kets in the Yamal-Nenets Autonomous Region (YNAR; northwestern Siberia)	35	1992–2006	JX090647, JX090675-JX090679, JX090688-JX090702, JX090711-JX090718, JX125366-JX125368, JX125382-JX125384	[36]
Taimyr: combined group of Dolgans and Nganasans in the Dudinsky District (northernmost Siberia, Taimyr Peninsula)	32	2000	JX090626-JX090632, JX090638-JX090646, JX090703-JX090710, JX125370-JX125375, JX125381, JX125386	[25]
Buryats: Irkutsk Region, Nukutsky and Alarsky Districts (southeastern Siberia)	35	2005, 2006	JX090605-JX090625, JX090680- JX090687, JX125364, JX125365, JX125369, JX125376, JX125380, JX125385	[36]
Yakuts: Sakha Republic (or Yakutia) (eastern Siberia)	17	2004–2006	Author’s numbers (yakXX, see Appendix A)	[35]
	15	1997	AY653781, AY653782, AY653787-AY653789, AY653796, AY653799, AY653828, EU594390-EU594395, EU594433	[37,38]
	35	2014	KM212957, KP143742-KP143745, KP165597-KP165605, KP184495-KP184499, KP202936-KP202945, KP230541, KT962021-KT962025	[39]
	38	2008–2019	OK143470, OK143474, OK143477, OK143478, OK143480, OK143482, OK143485, OK143489 OK143490, OK143494-OK143496, OK143498, OM025238-OM025253, author’s numbers (88–732, see Appendix A)	[40]
Chukchi: Chukotka Autonomous Region (northeastern-most Siberia)	123	1997–2008	Author’s numbers (chXX, schXX, see Appendix A)	[35]
Urban populations of Russia and western republics of the former USSR (listed from west to east), n = 179				
Estonia	21	2003	EU594383-EU594386, EU594400-EU594414, EU594434, EU594435	[37,38]
Latvia	33	2003–2005	AY603447-AY603466, AY653829, AY653858, AY653885, AY653886, EU594387, EU594388, JX096952-JX096958	[37,41]; Silamikelis & Legzdina, 2012
Belarus	31	2005	EU414067, EU414075, EU414077, EU414078, EU414080-EU414085, EU414087, EU414090, EU414091, EU414093, EU414095-EU414097, EU414107-EU414110, EU414114-EU414122, EU414124-EU414129, EU414132-EU414143	[42]
Russia (region not specified)	2	2004	AB205127, AB205128	Nakajima et al., 2005
Saint Petersburg	2	2003, 2014	AY509974, KT963508	Morozov, 2004; Kalinina et al., 2016
Moscow	5	2003, 2004	AB126580, AB126581, AY653786, AY653801, AY653847	Abe & Tran, 2003; [38]
Volgograd city (southern European Russia)	15	2003, 2004	AY653776, AY653780, AY653791, AY653798, AY653800, AY653805, AY653808, AY653809, AY653825, AY653826, AY653830, AY653839, AY653843, EU594382, EU594389	[37,38]
Tyumen city (eastern Ural)	2	2004	AY653871, AY653894	[38]
Kemerovo city (western Siberia)	8	2003, 2004	AY653821, AY653859, AY653862, AY653866, AY653873, EU594415-EU594417	[37,38]
Krasnoyarsk (southern Siberia)	11	2003, 2004	AY653810, AY653823, AY653824, AY653831, AY653870, AY653892, EU594429, EU594430-EU594432, EU594436	[37,38]
Irkutsk Region (southeastern Siberia)	8	2005	JX090648-JX090655	[36]
Chita city (eastern Siberia)	7	2004	AY653806, AY653811, AY653834, AY653849, AY653856, EU594398, EU594399	[38]
Khabarovsk city (far eastern Russia)	34	2003, 2004, 2015	AY653822, AY653833, AY653850, AY653855, AY653876 AY653878, EU594418-EU594427, KX925286-KX925303	[37,38]; Kotova et al., 2016
Central Asian Republics of the Former USSR and Mongolia, n = 50				
Uzbekistan	19	2003, 2007	AB222707-AB222715, AY653777, AY653814-AY653817, AY653819, AY653827, AY653832, AY653838, EU594397	[37,43]
Kazakhstan	2	2003	EU594396, EU594428	[38]
Tajikistan	1	2003	AY738889	[17]
Mongolia	28	2005–2006	AB263404-AB263412, AB270534-AB270550, DQ111986, DQ111987	[44,45]
Other Strains				
Prototype HBV strains with known (sub)genotypes from other parts of the world	319	1984–2015	See Appendix A for the accession numbers, locations, and years of collection	-
HBV strains from ancient tombs located in different parts of the world (including modern Russian and Central Asian territories)	124	up to 10,000 years ago	Author’s names; see the Appendix A for the names, locations and years of burials	[31]

**Figure 1 viruses-14-02465-f001:**
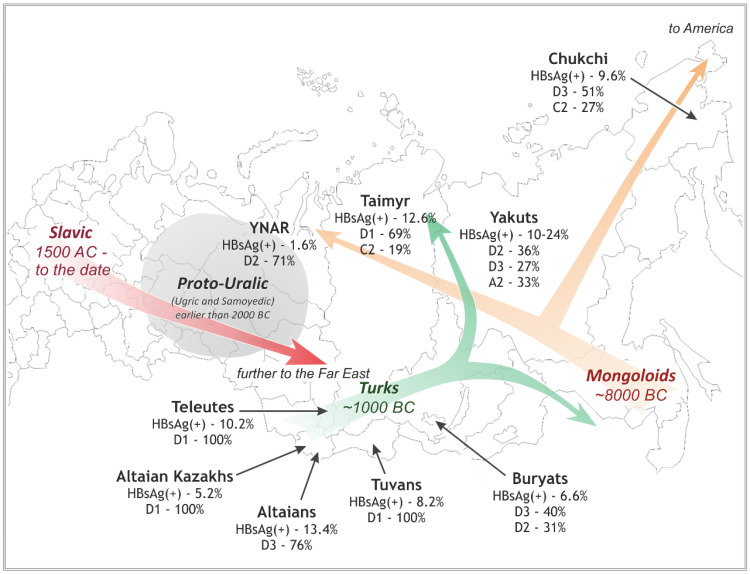
A schematic map showing the locations of the studied native Siberian populations. The prevalent HBV subgenotypes in these populations are shown along with reported HBsAg incidence (detailed data are in Table 2). The colored arrows designate the main migration pathways during the settlement of Siberia in the past ten millennia (see Section 4).

## 3. Results

### 3.1. HBV Viral Populations and Study Design

The main object of the study was to analyze HBV strains in nine geographical locations in Siberia that are settled mostly by indigenous people of different ethnic and language groups. These nine native groups and the regions in which they live are depicted in Figure 1. In total, 381 HBV strains from native populations were analyzed. A list of these and all other sequences studied in this paper is presented with comments and references in Table 1. 

Because today Siberian natives live in close contact with Russian-speaking populations (especially in regional capital towns), it is important to know whether HBV populations are common to both the aboriginal and the Russian-speaking populations (hereafter, we will say “Russian populations” to indicate historically newer settlers in Siberia). To answer this question, the study included 179 HBV strains sampled in large cities of Russia, Belarus, and the Baltic states (Table 1). We combined these strains into one reference group because the homogeneity of the HBV samples in urban populations of the Russian Federation and contiguous countries has been reported previously by a number of authors [35,37,38,42]. 

Many Siberian natives are believed to have descended from ancient Turkic-speaking tribes that migrated to Siberia from Central Asian territories long ago (see below). Because of this, we added a group of 50 HBV strains derived from modern-day Central Asian countries: Uzbekistan and Kazakhstan (which are Turkic-speaking nations), as well as Tajikistan and Mongolia (Table 1). We decided to include the later country in the "Central Asian" group because Mongolian people have a very long and rich history of relations with both the Russian population and the Siberian and Turkic peoples.

Finally, we included in the analysis 124 HBV sequences (Table 1) that have been recently collected in archeological tombs in different parts of the world (including modern Russian and Central Asian territories) [31] to determine whether the descendants of these ancient variants survive in present-day Siberia. 

All the listed HBV strains, along with 319 prototype sequences from GenBank with known (sub)genotypes, were used in a single phylogenetic analysis using the ML method with the aLRT-SH test for calculating branch support indices [33,34] (see Section 2). The resulting tree, consisting of 1053 HBV strains, is presented, due to its large size, in Appendix A. In this complete tree, one can find a branch of any studied strain, its code or accession number, the region and year of collection, and the ethnicity of the donor (Siberian strains). The overall layout of the tree, including the main regional and ethnic clusters, support indices for these clusters, topology, and relations among clusters inside the clades of HBV subgenotypes, is depicted in Figure 2.

The sequences included in the same analysis were of different lengths (681–3200 nt, see Section 2) as they were collected in different studies and by different authors (Table 1). As the Maximum Likelihood method is applicable for aligned sequences of different lengths [34], we used HBV sequences from 681 to 3200 nt in length in the same analysis to obtain more accurate results of genotyping and to avoid possible bias due to the analysis of trimmed, shorter sequences.

The tree (Appendix A) was used to determine the subgenotype of all the studied HBV strains. Additionally, for each strain, the HBsAg subtype was transcribed based on the sequence of its S-gene. The prevalence of the different subgenotypes and subtypes in all the studied groups is summarized in Table 2. This paper is chiefly concerned with comparing the studied groups by proportions of different HBV genotypes and subtypes. From this, conclusions are drawn whether the populations are epidemiologically homogeneous (i.e., a common population of HBV circulates within them), or if the HBV strains carried by populations are of different origins. At the end of the paper, authors speculate on the ways in which the current patterns of HBV genetic diversity in Siberia might have formed.

**Figure 2 viruses-14-02465-f002:**
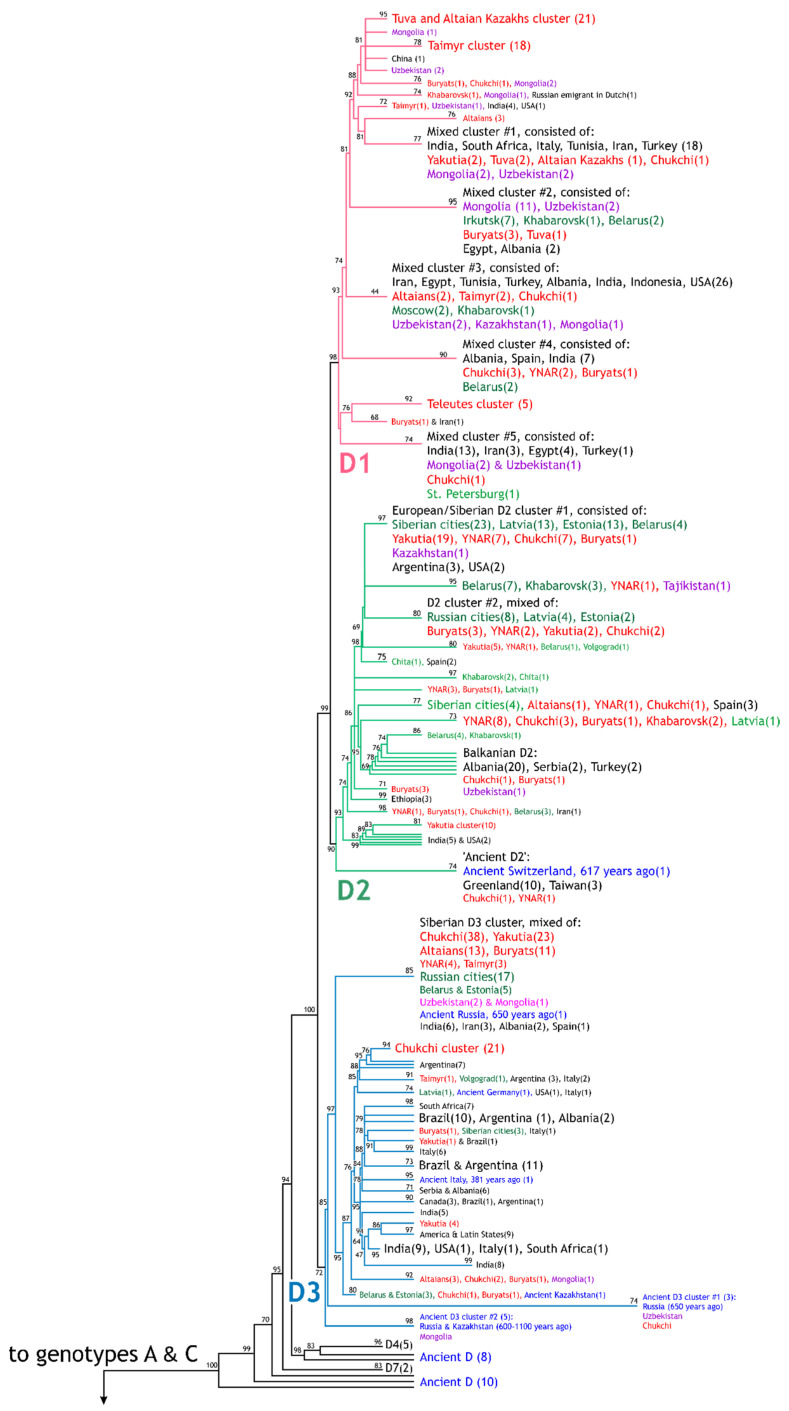
Schematic topology of the constructed phylogenetic tree (genotypes D, A–C, E–H). Clusters and strains are marked in: red: Siberian natives; green: urban populations of Russia, Belarus, and the Baltic States (“Russian population”); violet: Central Asian republics and Mongolia; blue: ancient samples from [31]. Strain numbers in the clusters are shown in brackets. Supporting indices, which were calculated using an aLRT-SH-like procedure (see Section 2), are shown near the corresponding clusters and branches. For a detailed tree of all 1053 samples, see Appendix A.

### 3.2. Differences in the HBV Subgenotypes in Siberian Natives and Urban Populations of the Former USSR

The first important conclusion that can be drawn from data in Table 2 (bottom rows) is that the ratio of HBV subgenotypes differs significantly in urban populations of Russia, Belarus, and the Baltic states; in the Central Asian republics; and in Siberian native populations, even when samples from the aboriginal groups are combined together into a synthetic cohort (of course, the distinct aboriginal groups differ from each other, which will be discussed below). Indeed, the main HBV subgenotype in the “Russian population” samples was D2 (56%; Table 2), and its prevalence here was reliably higher than in either the Central Asian republic samples (6%; *p* < 0.001) or Siberian native samples (24%; *p* < 0.001). By contrast, in the Central Asia and Mongolia group, the main subgenotype was D1 (62%; Table 2), which was significantly higher than in the “Russian population” (9%; *p* < 0.001) and the Siberian group (19%; *p* < 0.001). Finally, in samples from Siberian natives, all the three D subgenotypes were present in even proportions (Table 2), but the most common subgenotype was D3 (34%), and its prevalence was higher than in the “Russian population” samples (17%; *p* < 0.001) and the Central Asian samples (14%; *p* < 0.01). All this suggests that the aboriginal population of Siberia is still epidemiologically isolated from both the European population and the Turkic and Mongolian populations of Central Asia, even though these peoples have lived together and interacted actively in the same vast territory for at least the last thousand years. Furthermore, we will see that the native groups of Siberia are also very different from each other in terms of the HBV variants they host.

**Table 2 viruses-14-02465-t002:** Prevalence of the HBV subgenotypes and HBsAg subtypes in the studied groups. The most prevalent variants in each group are marked by grey fill.

Group	HBsAg (+) Rate in the Group ^1^	N	HBV Subgenotypes					HBsAg Subtypes					
			D1	D2	D3	A2	C2	ayw2	ayw3	ayw4	adw2	adrq+	N/D
Altaians	1.4%	21	4 19%	1 5%	16 76%	0	0	20 95%	1 5%	0	0	0	0
Altaian Kazakhs	5.2%	7	7 100%	0	0	0	0	7 100%	0	0	0	0	0
Tuvans	8.2%	18	18 100%	0	0	0	0	18 100%	0	0	0	0	0
Teleutes	10.2%	5	5 100%	0	0	0	0	4 80%	0	1 20%	0	0	0
YNAR	0–1.6%	35	2 6%	25 71%	5 14%	2 6%	1 3%	9 26%	21 60%	1 3%	3 8%	1 3%	0
Taimyr	12.6%	32	22 69%	0	4 12%	0	6 19%	25 78%	0	1 3%	0	6 19%	0
Buryats	6.6%	35	6 17%	11 31%	14 40%	1 3%	3 9%	17 48%	13 37%	1 3%	2 6%	2 6%	0
Yakuts	10.4–23.8%	105	2 2%	38 36%	28 27%	35 33%	2 2%	26 25%	39 37%	2 2%	35 33%	2 2%	1 1%
Chukchi	9.6%	123	7 6%	17 14%	63 51%	3 2%	33 27%	70 57%	16 13%	0	3 2%	33 27%	1 1%
Total: Siberian native populations	4.4%	381	73 19%	92 24%	130 34%	41 11%	45 12%	196 51%	90 24%	6 2%	43 11%	44 11%	2 1%
Urban populations of the former western USSR	2–4% (estimation)	179	17 9%	100 56%	30 17%	31 17%	1 1%	47 26%	87 49%	8 4%	33 18%	1 1%	3 2%
Central Asian Republics of the former USSR and Mongolia	4.1–6.2%; up to 16% (Mongolia)	50	31 62%	3 6%	7 14%	5 10%	4 8%	39 78%	4 8%	0	5 10%	2 4%	0

^1^ The rates of HBsAg positives in each group are indicated according to the reported data for Altaians [46], Kazakhs [47], Tuvans [48], Teleuts and Taimyr [25], YNAR [49,50,51], Buryats [36], Yakuts [52,53], Chukchi [35], across Siberia [25], urban population of the western former USSR [35,38,54,55,56], Central Asian Republics as cited in the systematic review [57], and Mongolia [58].

### 3.3. Diversity of the HBV Characteristics and Variants in Different Native Groups in Siberia

Nine native Siberian groups were studied (Table 1 and Table 2, Figure 1): in southwestern Siberia, Altaians, Kazakhs (who live in Altai Republic), Tuvans, and Teleuts; in northern Siberia, a combined group of the inhabitants of Yamalo-Nenets Autonomous Region (YNAR), consisting of the Nenets, Khants, Komi, Kets, and Selkups, and a combined group of the inhabitants of the Taimyr Peninsula, including the Nganasans and Dolgans; and in eastern Siberia, Buryats, Yakuts, and Chukchi. Table 2 presents the number of examined samples from each group, the reported estimation of chronic hepatitis B prevalence, and the calculated distribution of the HBV subgenotypes and HBsAg subtypes in each of the listed groups. 

The incidence of HBsAg-positive carriers in Siberian populations has been reported by different authors (see the comments in Table 2 for the citations), which have shown that the studied groups vary greatly in terms of HBV prevalence. Some groups may be considered as low-endemic for HBV B (e.g., YNAR [0–1.6% HBV prevalence]), but most are highly endemic groups according to the classification of [59], with 8–13% HBV prevalence in Tuvans, Chukchi, Teleuts, Taimyr Peninsula inhabitants, and Altaians and 24% prevalence in Yakuts (Table 2). For comparison, in the general population of Russia, the prevalence of HBsAg carriers was estimated to be 2–4% in the 2000s [56] (also see the references to Table 2). It is still unknown why many Siberian native groups demonstrate such high levels of HBV prevalence and which behavioral or other risk factors lead to the wide transmission of the virus, since no sound epidemiological study has been performed among these groups. Some studies [35,36,53] show that intravenous drug use is not common among any of the Siberian indigenous populations and thus does not strongly contribute to HBV transmission. Additionally, the incidence of HBsAg-positive carriers increases with age in some groups (Kets, Teleuts, Tuvans, Yakuts, YNAR, and Taimyr inhabitants) [25,50,60,61], indirectly indicating that existing (but not yet identified) risk factors affect the whole population rather than only specific cohorts with risky behavior.

The differences in incidence of HBsAg prevalence became the first evidence for the epidemiological heterogeneity of the studied groups. Analyzing the distribution of HBV subgenotypes also showed distinct viral populations circulating in Siberian native populations, even those with close geographical locations or common ethnic origin (see Figure 1 and Appendix A). 

#### 3.3.1. Southwestern Siberia: D3 (ayw2) in Altaians v. D1 (ayw2) in Kazakhs, Tuvans, and Teleuts

In southwestern Siberia, the most prevalent HBV subgenotype in Altaians was D3 (76%) (Table 2). The Altaians are a Turkic-language ethnic group (see Appendix A), with a current population of ~70,000 (here and below, population data are given according the Report of the Russian Census of 2010 [62]), and they live mostly in Altai Republic (Figure 1). At the same time, another subgenotype, D1, was absolutely dominant in the small group of Teleuts (100%, Table 2), who are close relatives of the Altaians and are even known as the sub-ethnos of “Upper Altaians”, living less than 500 km north of the Altai Republic in the Kemerovo Region (see Table 1 and Figure 1). Moreover, D1 was the main HBV subgenotype in other Turkic-language groups of Siberian Kazakhs (100%, Table 2), who live in the closest district to the Altaian settlements in the same republic (distance between the Altaians and Kazakhs locations is about 300 km, Table 1, Figure 1). Finally, the same D1 dominated among the Tuvans (100%, Table 2), who represent another large (more than 260,000 people) Turkic-language group in Siberia. The Tuvans live in Tuva Republic, which shares a border with the Altai Republic. Thus, Tuvans are located rather closely to Altaians and Kazakhs, who are ~500 km to the west. 

Thus, in three of these four studied groups of people, which are ethnically and geographically close to each other (see Appendix A), D1 was dominant, while D3 was dominant in the fourth. The differences in the incidence of D1 and D3 among the group of Altaians, Kazakhs, Teleuts, and Tuvans were statistically significant according to all pair comparisons between them (*p* < 0.05–0.001). This supports the existence of some yet unknown epidemiological barriers which isolate the Altaians (with predominance of the D3) from the surrounding HBV (D1) circulating in the closest populations.

#### 3.3.2. Northwestern and Northern Siberia: D2 (ayw3) in YNAR v. D1 (ayw2) in Taimyr

In northwestern Siberia, we examined the YNAR (Table 1 and Table 2), a multi-ethnic region at the north of the Ural Mountains (Figure 1) with extremely low population density; a total of 0.5 million people (including Russian urban populations, which cover about 83% of the general population of the region) lives in a territory of 0.77 million square kilometers. Among the non-urban indigenous populations, the most numerous are the Nenets (29,800 people in YNAR), Khants (9500), Komi (5150), and Selkups (2000). All these groups belong to peoples of the Uralic language family; the Komi are the members of the Finno-Ugric group, while Nenets, Khants, and Selkups represent different branches of the Samoyedic subfamily of peoples who are a result of intermixing between the western Ugric tribes and the ancient Siberian Mongoloid tribes. For additional data about language, genetic, and anthropological classifications of Siberian natives, see Appendix A. Another northwestern Siberian group, the Kets, is very different from the others, descending from the ancient Yeniseian peoples that have almost disappeared and having proposed common ancestry with Native American groups. 

Previous studies [25,36,50] have shown that the HBV types circulating in all the ethnic groups in YNAR, including the groups in our study (Figure 1, Table 1), are homogenous and do not demonstrate any statistically significant differences. Because of this, we considered these peoples as a single epidemiological group of the YNAR, as shown in Table 1 and Table 2. 

As mentioned above, the aboriginal peoples of YNAR represent a unique group because of the extremely low HBV endemicity (<1.6%; Table 2). In terms of HBV epidemiology, the inhabitants of YNAR are more similar to urban “Russian populations” in our study than to any other aboriginal groups located anywhere in Siberia (see Table 2). Like the “Russian population,” the peoples in YNAR demonstrated low presence of HBsAg carriers and predominance of the D2 subgenotype (71%), and, consequently, subtype ayw3 (60%; Table 2); the association between ayw3 and D2, as well as ayw2 and D1/D3, is well known [17]. Thus, the YNAR group statistically significantly differs from all other Siberian groups and the groups of Central Asia (Table 2) in incidences of D2 and ayw3 (*p* < 0.05–0.001) but is similar to the “Russian population,” in which D2 prevalence is 56% (Table 2). 

In our study, two other groups had notable prevalence of D2: the Yakuts (36% D2) and Buryats (31%; Table 2), although the prevalence of D2 in the YNAR (71%) was statistically higher (*p* < 0.001, *p* < 0.002). Figure 2 shows that the HBV strains in the D2 clade were intermixed regardless of the population from which they were collected—Russian, Yakuts, Buryats, or YNAR. This suggests that the D2 HBV population in these Siberian and “Russian” groups is common, despite large geographical distances between territories of the YNAR, Yakutia (Sakha Republic), and Buryatia (2000–3000 km, Figure 1). 

In another north-Arctic Siberian region, the Taimyr Peninsula (Figure 1), two main indigenous groups were studied, the Nganasans and Dolgans (Table 1). They have different ethnic origin, but now live alongside each other in the same settlements. The Nganasans, having only about 860 representatives, belong to the above-mentioned Uralic Samoyedic language family, while the more numerous group, the Dolgans (about 5500 persons on Taimyr), is a Turkic-language group that arrived to the region later (19th century) and is in fact a northwestern branch of the Yakuts (see below). Previous studies showed no differences in the circulating HBV types between Dolgans and Nganasans [25]; thus, we combined them into one group in our study. In Taimyr, the dominant HBV subgenotype was D1 (69%, Table 2), which, surprisingly made these populations similar to those located in southwestern Siberia (Kazakhs, Tuvans, Teleuts), but distinguished it from populations with D2 or D3 prevalence (Altaians, YNAR, Yakuts, Buryats, Chukchi) (differences confirmed statistically; *p* < 0.05–0.001). Interestingly, in this study, a significant incidence of HBV subgenotype C2 (19%, Table 2, Figure 2) was first discovered in the Taimyr. Genotype C is common mostly in southeastern Asia [17,20], and had never been previously reported so far north in Arctic Siberia (the Taimyr Peninsula is northernmost point of continental Eurasia). Since HBV genotype C has important implications in terms of clinical prognosis and therapy tactics, causing more severe infection with higher risk of cancer development than genotype D (cited by the review [63]), the discovered high prevalence of genotype C in Taimyr and Chukotka (see Figure 2 and below) should be taken into account by physicians and public health specialists in Siberia.

#### 3.3.3. Eastern Siberia: D3 (ayw2) in Buryats, D2 (ayw2) and A2 (adw2) in Yakuts, and D3 (awy2) and C2 (adrq+) in Chukchi

In southeastern Siberia, the large group of the Buryats was studied (Figure 1, Table 1). The Buryats (population in Russia ~460,000) belong to a numerous Mongolic-language family and are traditionally considered as a northern sub-ethnos of Mongolians. In this group, all the HBV subgenotypes—D1, D2, and D3—were circulating simultaneously with 17%, 31%, and 40% prevalence, respectively (Table 2). This absence of statistical differences in subgenotype rates makes the Buryats similar to the synthetic group of “Siberian natives” (with 19%, 24%, and 34% prevalence of D1, D2, and D3; Table 2). This may suggest that the Buryats are not isolated from surrounded aboriginal peoples and probably exchange HBV variants with other Siberian populations.

In eastern Siberia, the population of Sakha (Yakutia) Republic citizens was studied (Figure 1, Table 1 and Table 2). The Yakuts are the biggest non-Russian ethnic group in eastern Siberia, with a population of nearly half a million. The Yakuts are not considered “true” aboriginal natives, since they migrated into their present-day territory only ~400 years ago. As the eastern-most ethnic group of the Turkic-language family in the world, the Yakuts possess pronounced Mongoloid phenotypical traits while intermixing with ancient Mongoloid Tungus tribes that formerly inhabited the Baikal region and Lena River basin. 

Prevalence of the HBV subgenotype D2 in the Yakut population (36%, Table 2) makes these people statistically similar to the Buryats (31% D2 prevalence). In addition, as mentioned above, D2 was the main subgenotype in YNAR (71%; Table 2). However, the highest prevalence of HBsAg carriers (up to 24% according to [52]) and significant incidence of HBV genotype A (33%; Table 2) makes the Yakuts sharply different not only from the YNAR population, but from any other Siberian group in our study. 

Finally, at the far northeastern end of Siberia, the group of Chukchi was studied (Table 1). The Chukchi represent a relatively numerous (about 16,000 population) and very ancient aboriginal group of the Chukotka peninsula (Figure 1), belonging to the unique Kamchatkan family of peoples. The Paleo-Mongoloid ancestors of the Chukchi apparently moved to Chukotka over 5000–6000 years ago. The anthropological and genetic origins of the Chukchi give evidence to a common ancestor between them and Native Americans. Among Chukchi, the main HBV subgenotype was D3 (51%), similarly to the eastern Buryats (40%) and even to the Altaians (76%; Table 2). The main feature of this group, unusual for Russia, is the high incidence of subgenotype C2 (27%), which makes the Chukchi different from all the other Siberian groups, except the Taimyr inhabitants (19%; Table 2). Despite the large geographical distance between the Chukotka and Taimyr peninsulas (over 3000 km), all the Siberian strains of the C subgenotype in this study clustered into a joined clade of these two regions with good statistical support (0.86; Figure 2), indicating the common origin of HBV circulating among these groups of people. So, there might exist (or have previously existed) a mode of HBV transmission along the shore of the Arctic Ocean.

## 4. Discussion

As demonstrated above, many different HBV types circulate in Siberia, with distinct aboriginal groups often carrying different subgenotypes. But how did this pattern of HBV genetic diversity emerge, and, more widely, how might HBV have invaded Siberia?

HBV is known to have historically arisen not in Siberia, but in much more southern regions of the world, probably North Africa or the Middle East [17]. Thus, it should be assumed that HBV was introduced into Siberia along with the migrations of various populations in the past.

The first known *Homo* to live in Siberia occupied the region in the Paleolithic era, at least 45,000 years ago. These ancient inhabitants apparently disappeared and did not leave any known modern descendants. Much later, several big waves of settlement in Siberia occurred, which can be described very simplistically as follows (Figure 1) [64].

Since the Neolithic era, the most ancient known populations in eastern Siberia were Paleo-Mongoloids, who migrated there 8000–10,000 years ago. The modern descendants of these Mongoloid tribes are the Chukchi and possibly the Kets (see Table 1 and Appendix A). Of course, there are other ethnic groups of Mongoloid origin in Siberia, including Koryaks, Evenks (Tungusy), and others, but here we discuss only the groups that we studied. It is believed that the Siberian Paleo-Mongoloids had also given rise to Native Americans. 

Western Siberia was settled by the Uralic tribes at least 4000 years ago. These Uralic peoples divided into two major branches, giving origin to the Finno-Ugric group of peoples that further spread into the eastern part of Europe, and the Samoyedic group of peoples that occupied the territory of northern and central Siberia. Among modern Siberian aborigines, Komi are descendants of the Ugric people, while Samoyedic descendants include Khants, Nenets, Selkups, Nganasans, and some others, not included in our study. 

Approximately 3000 years ago, the Turkic tribes started active migrations to southern Siberia, eventually also moving to the north (Figure 1). Intermixing between the Turkic people and Siberian Mongoloids, the Turkic people gave rise to many modern aboriginal groups, including the Altaians, Teleuts, Tuvans, and Kazakhs in the south, and even the Yakuts and Dolgans in the very northeast of Siberia. The Turkic migrations in Siberia took a long time; the formation of the Yakuts nation had only been completed in the 17th century. Late in this period, during the 12th and 13th centuries, the Mongols, one of the descendants of these ancient Turkic tribes (despite of the name, the Mongols are not genetically Mongoloid, originating, according to the most popular hypothesis, from Xiongnu, or Hunnu, people), moved back to the West, sweeping through all of Siberia and Eastern Europe with the Genghis Khan conquest. The closest Siberian branch of these historical Mongol tribes now are the Buryats. 

In the 16th century, Russians began their active conquest of Siberia, which generally continues to this day. Certainly, this massive expansion hugely influenced all the smaller Siberian tribes. Due to these historical migration processes, at least 40 aboriginal groups exist in Siberia today [62], differing greatly in their origin, language, and traditions. Modern genetic and linguistic classification of Siberian natives is very complicated and controversial (see Appendix A). However, as we saw above, overall, four main waves of migration formed the existing diversity of the indigenous peoples of Siberia: Mongoloid, Uralic (Ugric and Samoyedic), Turkic, and European Slavic (Russian) (Figure 1). This may explain why the genetic HBV types collected in Siberia are highly diverse. 

It may be assumed that the C2 subgenotype of HBV was introduced into Siberia by the Mongoloids, since genotype C, as we know it, is one of the main HBV types in southeastern Asia, the territory in which Mongoloid peoples originated [17,20]. Surprisingly, HBV genotype C, being rare in the majority of studied populations, appeared to be prevalent in two geographically remote populations, in Chukchi and in Taimyr. Although we could not find any evidences of relationship or other connections between these two populations, there are two possible explanations for such unexpected similarity in genotype C prevalence. Firstly, this HBV genotype could be imported to both populations through Yakutia via the ancient route of migration. However, genotype C is not prevalent in modern Yakutia, its prevalence is only 2% compared to 19% on Taimyr and 27% on Chukotka. Secondly, genotype C could be introduced from Chukotka to Taimyr in modern times through seaports at the Arctic shore that have been operating since the beginning of the 20th century. Theoretically, some HBV strain(s) of the genotype C might have been imported to Taimyr with sailors and then spread there due to a “founder effect”.

Subgenotype D1 was obviously brought into Siberia with the Turkic migrations, since, in the modern world, this subgenotype is most common in historical homelands of the Turkic tribes [65]: Afghanistan [66], Iran and the Persian Gulf [67,68], and Turkey [69,70,71]. In addition, D1 is the main subgenotype among Turkic populations of the Central Asian republics and Mongolia (Table 2). We can thus hypothesize that the introduction of D1 into Siberia occurred enough times to form distinct phylogenetic clusters, which represents HBV subvariants of the Tuvans and Kazakhs, Teleuts, and Taimyr Peninsula inhabitants (see the upper part of Figure 2).

Subgenotypes D2 and A2 most probably entered Siberia with the Russian expansion, as these HBV types are the most common in Russia and in Eastern European countries (Table 2) [35,37,38,42]. Figure 2 shows that no clusters formed within D2 and A2 clades according to geographical region or ethnicity of the hosts; rather, strains from urban Russian and Eastern European populations are intermixed with strains from Siberian natives. It may be even suggested that the incoming Russians were the main source of HBV for the native populations of YNAR (where 71% of the found strains were D2) and Yakutia (36% D2 and 33% A2, Table 2). However, this does not explain why the incidence of HBsAg carriers is so high (up to 24%) in Yakutia and so low in YNAR (1.6%) (Table 2); why the prevalence of the A2 among the Yakuts is twice as high as among the “Russian population” (33% vs. 17%, *p* < 0.005); and, finally, why the Dolgans (Taimyr Peninsula people who are the northern relatives of Yakuts) do not carry D2 and A2 (Table 2) but have significant incidence of subgenotype C2 (19%), which is very rare in Yakuts (2%, *p* < 0.005). All these questions remain to be answered in further investigations. For now, we can suppose that, apart from epidemiological factors in HBV spread in Siberia, there were a number of “bottle-necks” when a specific HBV type developed an advantage under yet unknown circumstances.

It is not clear how D3 entered and spread in Siberia. In the modern world, there are no regions with a clear predominance of this HBV subgenotype. As a minor subgenotype, D3 has been reported in South America [72,73], India [74,75], and Egypt [76]. It seems that the Siberian aboriginal population in which D3 is the most prevalent subgenotype (34%, Table 2) is the main reservoir of D3 on the planet. Moreover, we can conclude that there exists a Siberian variant of D3, strains of which have formed a large and explicit cluster inside D3 (see the upper part of the D3 clade in Figure 2; support index is 0.85). Of course, as mentioned above, it is unlikely that D3 arose in Siberia. Most probably, it had come from more southern regions (e.g., India, where it circulates to date [74,75]) but became established in Siberia, having been displaced from southern countries by the more modern subtype D1 [17,65]. In turn, from Siberia, D3 may have entered South America moving with the ancestors of Native Americans. More recently, it may have returned to Europe with migrants from Latin America to the Mediterranean, where this subtype is also found today (see D3 clade in Figure 2 and the full tree in Appendix A). In addition, the D3 clade in Figure 2 includes some ancient strains from Russia and Kazakhstan [31], supporting the hypothesis that this subgenotype was already common in those regions (as well as Siberia) 600–1100 years ago. 

We clearly understand that the above speculations are completely hypothetical. In this paper, we do not provide scientific evidence to suggest that HBV spread in Siberia exactly as we suppose. However, our reasonings do not contradict the static picture of the genetic diversity of HBV in Siberia that we have described. We deliberately did not use any phylodynamic or phylogeographic methods in this study because, to date, we are unable to reasonably choose one of the existing models of the HBV molecular clock and believe that the reported rates of mutations in the HBV genome are still debatable. In addition to that, we realize that the phylogenetic methods we used, and especially the fact that we included HBV sequences of different lengths in the same analyses (including some sequences represented only by the small S-gene), are suitable for subgenotyping purposes but are unable to disclose precise phylogenetic relationships between the strains. Because of this, we publish all the sequences in Appendix A and invite more sophisticated researchers to analyze them with better methods after developing a commonly recognized model for HBV evolution.

## Data Availability

The data presented in this study are available in Appendix A. Any additional data are available on request from the corresponding author: victormanuilov@medipaltech.ru.

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
