# Peer review of "Genetic Diversity and Possible Origins of the Hepatitis B Virus in Siberian Natives"

_viruses, 2022, doi:10.3390/v14112465_

Round 1
Reviewer 1 Report
In this study, Manuylov et al. took advantage of the epidemiologically isolated nine aboriginal Siberian populations (Altaians, Kazakhs, Tuvans, Teleuts and Taimyr, YNAR, Buryats, Yakuts, Chukchi) and performed a comprehensive assay about the HBV genetic diversity and possible origins in Siberian natives by enrolling 381 HBV sequences, 179 HBV sequences from different urban Russian populations, 50 HBV sequences from Central Asian, 124 HBV sequences from ancient tombs worldwide, and 319 prototype HBV sequences for genotyping reference. Through this, the authors wanted to give an answer/hypothesis of how HBV historically spread throughout Siberia. This is a well-done study and the data is clearly presented.
General comments
1. One of the concerns about this study is the authors included variable lengths of HBV sequences into a same phylogenetic assay which may lead to some bias in the genotyping classification. Please explain it carefully.
2. For gtC, which is most predominant in southeastern Asia but found in Chukchi and Taimyr populations in Siberian, is there some relationship between these two aboriginal people?
3. Is it a normal/common style to present the statistical significance as “p < 0.05–0.001” (line 294; 325-326;…)?
4. some expressions need to be rephrased: “That means that the” line 294; “no known” line 404
Author Response
Dear Reviewer 1,
we appreciate you a lot for reviewing our paper "Genetic diversity and possible origins of the hepatitis B virus in Siberian natives".
Here are point by point response to your valuable comments:
General comment: In this study, Manuylov et al. took advantage of the epidemiologically isolated nine aboriginal Siberian populations (Altaians, Kazakhs, Tuvans, Teleuts and Taimyr, YNAR, Buryats, Yakuts, Chukchi) and performed a comprehensive assay about the HBV genetic diversity and possible origins in Siberian natives by enrolling 381 HBV sequences, 179 HBV sequences from different urban Russian populations, 50 HBV sequences from Central Asian, 124 HBV sequences from ancient tombs worldwide, and 319 prototype HBV sequences for genotyping reference. Through this, the authors wanted to give an answer/hypothesis of how HBV historically spread throughout Siberia. This is a well-done study and the data is clearly presented.
Response: Thank you very much for the good appreciation of the manuscript and the study.
Comment 1: One of the concerns about this study is the authors included variable lengths of HBV sequences into a same phylogenetic assay which may lead to some bias in the genotyping classification. Please explain it carefully.
Response 1: Indeed, choosing the genomic region for the analysis may affect the (sub)genotyping results. For example, one may look at one of our previous trees constructed by the same method but using only the 681-nucleotide sequences of the HBV S-gene. This tree (please see the attached file) is based on 700 sequences, including 345 strains from Siberian natives, 200 from the former USSR cities and 155 prototype strains from GenBank.
Using this “short” 681-nt sequences we couldn’t determine the subgenotype for 8/345 (2.3%) Siberian strains, although for all of these eight sequences genotype D was clearly identified. Also, based on these 681-nt long sequences we were unable to discriminate the A1 and A2 subgenotypes. At the same time, it may be seen from the “big” tree in the Supplement A to the paper, that all newly analyzed HBV sequences (please refer to the alignment in Supplement B to the paper) can be assigned to a certain subgenotype. By this particular reason we strive to analyze as long sequences as possible to get more detailed results.
Maximum Likelihood method we used (Guindon et al., 2011) is applicable for analysis of aligned sequences of different lengths. It ignores missed sites when calculating genetic distances between strains, but takes into pairwise comparisons all the informative sites which match both sequences. Due to this, in the same analysis we may use sequences of different length. Our phylogenetic exercises suggest that analysis of an alignment which includes sequences of 681-3200 nt in length, gives more informative results comparing to an analysis of sequences that all have the same length of 681 nt (at least in the PhyML 3.0 online software we used, http://www.atgc-montpellier.fr/phyml). Due to this, we included variable lengths of HBV sequences into a same phylogenetic analysis to avoid any possible bias in genotyping results.
We added this point to the paper (lines 181-186 in revised manuscript).
Comment 2: For gtC, which is most predominant in southeastern Asia but found in Chukchi and Taimyr populations in Siberian, is there some relationship between these two aboriginal people?
Response 2: In the ethnographic and historical literature we could not find any evidences of relationship or other connections between these peoples that could contribute to virus transmission. The distance between Dudinka (the "capital" of Taimyr) and Anadyr city on the Chukotka peninsula is about 3600 kilometers along a geodetic line. It is impossible to move between these two points by land, as they are separated not only by the tundra but also by the impassable Putorana Plateau. The Chukchi are an ancient Mongoloid tribe and moved to their locations on the Chukotka long before the Nganasans and Dolgans (who are the indigenous citizens of the Taimyr) had arisen. The first settlements of the Chukchi are dated from IV-III thousand years BC, while Dolgans came to the Taimyr only in the 17th century. All these peoples belong to different genetic and language groups (please see the Table C-3 in the Supplement C to the main paper): the Chukchi, as mentioned, are Mongoloids of Kamchatkian family; Dolgans are recently arisen members of the Turkic-language family; and Nganasans belongs to the Samoyedic branch of the Uralic family.
We have only two hypotheses of how the HBV might have moved between the Chukotka and Taimyr. The first possible way is through South Yakutia. Today, there are several roads there and it is possible that the same route operated in the past. But, in this case we should see a significant level of genotype C in Yakutia, and we don’t: only 2% of strains from Yakutia were of genotype C, compared to 19% on Taimyr and 27% on Chukotka.
The second possible way lies through the Arctic ocean in modern times. The North Seaway existed since the beginning of the 20th century, but in 1960-70’s the Soviet Union started improve it, and now this seaway is used very actively. Dudinka and Anadyr are important seaports at the Arctic shore. Theoretically, some HBV strain(s) of the genotype C from Chukotka might have reached the Taimyr along with sailors, and then spread there due to a “founder effect”.
We discussed in briefly in the revised manuscript (lines 444-455).
Comment 3: Is it a normal/common style to present the statistical significance as “p < 0.05–0.001” (line 294; 325-326;…)
Response 3: We used this form of presentation in order not to list all the p-values in multiple paired comparisons, but simply to make it clear that all the differences in the paired comparisons were valid. But we agree that it looks unconventional. For example, the current phrase at the line 294:
“The differences in the incidence of D1 and D3 among the group of Altaians, Kazakhs, Teleuts, and Tuvans were statistically significant according to all pair comparisons between them (p < 0.05–0.001)”
must sound as:
“The prevalence of D3 in Altaians (76%) was significantly higher than in the groups of Tuvans (0%, p<0.001), Kazakhs (0%, p<0.005) and Teleutes (0%, p<0.05). Analogically, the prevalence of the D1 in the Tuvans, Kazakhs and Teleutes (100%) was significantly higher than in the Altaians (19%, p<0.001, p<0.001 and p<0.05, respectively)”.
The phrases at the lines 325-326 and 347 may be easily changed in the same way. The second variant looks more “scientific”, but more sophisticated for reading. If possible, we would ask Reviewer and the Editors suggest which variant is better and we will make the final corrections if needed.
Comment 4: Some expressions need to be rephrased: “That means that the” line 294; “no known” line 404
Response 4: Thank you for these remarks. The sentences are rephrased to:
(Line 294-297 in revised manuscript): This supports the existence of some yet unknown epidemiological barriers which isolate the Altaians (with predominance of the D3) from the surrounding HBV (D1) circulating in the closest populations.
(Line 405): These ancient inhabitants apparently disappeared and hadn't left any known modern descendants.
Best regards,
Victor Manuylov, PhD
October 27, 2022

Reviewer 2 Report
The biggest lacuna in the article is it seems more like a review than a research article. The same is very well acknowledged by the authors in lines 486-489. To summarize, a collection of sequences available from database has been analyzed for its phylogenetic relation and a hypothesis has been proposed about their movement from one geographical reagion to another. The differential presence and prevalence of HBV genotypes is well documented. Having said so, the crisp and clear presentation of the article must be acknowledged. The manuscript may be considered only if the authors dwell upon the genetic variations occuring between the various genotypes, as in their correrlation with respect to the phylogenetic analysis as well as implications on natural selection, if any.
Author Response
Dear Reviewer 2,
we appreciate you a lot for reviewing our paper "Genetic diversity and possible origins of the hepatitis B virus in Siberian natives".
Here are point by point response to your valuable comments:
Comment 1: The biggest lacuna in the article is it seems more like a review than a research article. The same is very well acknowledged by the authors in lines 486-489. To summarize, a collection of sequences available from database has been analyzed for its phylogenetic relation and a hypothesis has been proposed about their movement from one geographical region to another.
Response 1: We consider this article not as a review or a meta-study, but as an original research work because all the Siberian sequences we used had been obtained originally in the field with direct involvement of the authors. Moreover, the vast majority of these sequences and related data have never been published in English-language peer-reviewed journals or elsewhere.
Indeed, the main results of the study were gained using 381 HBV strains isolated from Siberian natives. These strains include (please see the Table 1 in the main text, and the references to the table):
21 strains of the Altaians, 7 strains of Kazakhs, 5 from Teleuts, 35 strains from YNAR, 35 of Buryats, 32 from Taimyr were collected in field by Ludmila Osipova’s group and primarily analyzed by Irina Netesova, Victor Manuylov, Ludmila Bezuglova and Elena Chub (published in Russian only: Manuilov et al., 2010, Manuilov et al., 2015, see the Reference list to the main paper);
18 strains from Tuva, 17 from Yakutia and 123 from Chukotka were collected and sequenced by the group of Vladimir Chulanov and Inga Karandashova, and have never been published previously even in Russian journals;
35 strains from Yakutia were collected and sequenced by the group of Yulia Ostankova, Alexander Semenov and Sergei Mukomolov, and have been published in Russian only (Semenov et al., 2016);
38 strains from the Yakuts were collected by the group of Karen Kyuregyan and Anastasia Karlsen (Karlsen et al., 2022);
Finally, 15 strains from Yakutia and more than 100 strains from city population of the former USSR used in the analysis were collected and sequenced by Tatjana Tallo and colleagues (Tallo et al., 2004; Tallo et al., 2008).
Besides this, the HBsAg prevalence was firstly reported in the Siberian groups of the Altaians and Kazakhs by (Netesova et al., 2001; Netesova et al., 2003); YNAR (Netesova et al., 2004; Manuilov et al., 2005); Teleuts, Buryats and Taimyr (Manuilov et al., 2010; Manuilov et al., 2015).
A comprehensive analysis of the HBV genotypes, subgenotypes and HBsAg subtypes among wide cohorts of Siberian natives has never been performed and published before. It has just been reported (sub)genotypes prevalence in a number of groups, like Altaians, Kazakhs, Buryats, Taimyr and YNAR (Manuilov et al., 2010; Manuilov et al., 2015) but it has been first shown in the present paper that many aboriginal groups of Siberia are epidemiologically isolated from each other and carry different HBV subgenotypes.
So, this study doesn’t fit the definition of review, since it uses own data instead of previously published. We do believe that data on HBV genotype distribution in epidemiologically isolated populations of Siberia natives are of interest for many virologists and are useful for understanding the epidemiology of hepatitis B.
Comment 2: The differential presence and prevalence of HBV genotypes is well documented. Having said so, the crisp and clear presentation of the article must be acknowledged. The manuscript may be considered only if the authors dwell upon the genetic variations occurring between the various genotypes, as in their correlation with respect to the phylogenetic analysis as well as implications on natural selection, if any.
Response 2: Yes, we do believe that distribution of the HBV subgenotypes and HBsAg subtypes in different Siberian native groups, documented for the first time in our paper, is the main and the most valuable result of our study. Previously the HBV genetic landscape in Russia has been concerned by many authors as relatively homogenous, as mostly city population and ethnic majorities have been studied. Now we have shown that not only “eastern European” D2 and A2 are circulating at this large territory, but also “Turkic/Middle Eastern” D1 and “Eastern Asian” C2, and, moreover, that the main subgenotype in Siberia is D3, which has been previously believed to be a minor subgenotype with no predominance elsewhere.
We looked for some genetic markers (signature amino acid or nucleotide substitution and/or patterns) in the studied strains, but couldn’t find any that would be reliably specific for a studied populations, but not subgenotype (or subtype) -specific in general. For this reason, we did not discuss any genetic features of the strains in the article.
Best regards,
Victor Manuylov, PhD
October 27, 2022
Round 2
Reviewer 2 Report
The authors have answered the queries satisfactorily and the paper may be accepted for publication.